# *In vivo* two-photon microscopic observation and ablation in deeper brain regions realized by modifications of excitation beam diameter and immersion liquid

Kazushi Yamaguchi[1,2,3☯], Ryoji Kitamura[1,2☯], Ryosuke Kawakami[1,2,4], Kohei Otomo[1,2,3,5,6], Tomomi Nemoto[1,2,3,5,6]*

1 Graduate School of Information Science and Technology, Hokkaido University, Sapporo, Hokkaido, Japan, 2 Research Institute for Electronic Science, Hokkaido University, Sapporo, Hokkaido, Japan, 3 Division of Biophotonics, National Institute for Physiological Sciences, National Institutes of Natural Sciences, Okazaki, Aichi, Japan, 4 Graduate School of Medicine, Ehime University, Toon, Ehime, Japan, 5 Exploratory Research Center on Life and Living Systems, National Institutes of Natural Sciences, Okazaki, Aichi, Japan, 6 Department of Physiological Sciences, The Graduate School for Advanced Study, Hayama, Japan

☯ These authors contributed equally to this work.
* tn@nips.ac.jp

**Data Availability Statement:** All relevant data are within the paper and its Supporting Information files.

## Abstract

*In vivo* two-photon microscopy utilizing a nonlinear optical process enables, in living mouse brains, not only the visualization of morphologies and functions of neural networks in deep regions but also their optical manipulation at targeted sites with high spatial precision. Because the two-photon excitation efficiency is proportional to the square of the photon density of the excitation laser light at the focal position, optical aberrations induced by specimens mainly limit the maximum depth of observations or that of manipulations in the microscopy. To increase the two-photon excitation efficiency, we developed a method for evaluating the focal volume in living mouse brains. With this method, we modified the beam diameter of the excitation laser light and the value of the refractive index in the immersion liquid to maximize the excitation photon density at the focal position. These two modifications allowed the successful visualization of the finer structures of hippocampal CA1 neurons, as well as the intracellular calcium dynamics in cortical layer V astrocytes, even with our conventional two-photon microscopy system. Furthermore, it enabled focal laser ablation dissection of both single apical and single basal dendrites of cortical layer V pyramidal neurons. These simple modifications would enable us to investigate the contributions of single cells or single dendrites to the functions of local cortical networks.

## Introduction

Brain functions such as senses, behaviors, intelligence and emotions are believed to be achieved by intercellular communications in either local or global neural networks. However, little is known about cellular mechanisms implementing such complex brain functions even in local neural networks [1]. To reveal the cellular mechanisms, investigations of dynamics in

**Funding:** The funders had no role in study design, data collection and analysis, decision to publish, or preparation of the manuscript.

**Competing interests:** No authors have competing interests.

neural networks with single-cell levels are necessary. In local neural networks, neural-ensemble activities and neuron-astrocyte interactions induce synaptic plasticity that controls the efficiency of intercellular communications [2–5].

*In vivo* two-photon excitation microscopy has subcellular spatial resolution and superior observation depth advantages for monitoring of synaptic plasticity and neural population activities [6]. On one hand, the high spatial resolution is achieved with the localization of the excitation area caused by direct proportionality between the two-photon excitation efficiency and the square of the excitation photon density. On the other hand, near-infrared excitations of visible fluorescent probes decrease absorption and scattering, resulting to superior penetration depth [7–9]. However, in conventional two-photon microscopy systems equipped with a generally used multi-alkaline photomultiplier tube (PMT) and a near-infrared Ti:Sapphire laser light source, observations of synaptic structures, somata or $Ca^{2+}$ activities were hardly performed in deeper layers than in cortical layer II/III, layer VI, or layer II/III, respectively [10–12]. In addition, the focal laser ablation for examining or manipulating functions of single neurons in local cortical networks was limited to cortical layer II/III at deepest [13–15]. These limitations of depth in observations and in focal laser ablations have prevented the investigation of the activities in various types of single neural cells.

Such limitations are mainly derived from the supralinear decline of the two-photon excitation efficiency in deeper regions of the specimens, which results from a decrease in the photon density [16] due to absorption, scattering, and optical aberrations. With this, the excitation laser power at the focus could be enhanced to increase the two-photon excitation efficiency. Previously, novel two-photon microscopy systems equipped with high-peak-power and long-wavelength laser light sources were employed to reduce absorption and scattering in living mouse brains [17, 18]. Moreover, optical aberrations of the focusing excitation laser light beam are usually induced by heterogeneous distributions of the refractive index (RI) in the specimen, preventing tighter focusing that expands the focal volume especially in deeper regions. To compensate for optical aberrations, sophisticated adaptive optical (AO) techniques have been proposed and applied for two-photon imaging of fixed or living mouse brains [19–23]. However, incompatibilities of such technologies with conventional two-photon microscopy have prevented their widespread use. In this study, we developed a method for evaluating focal volumes in living mouse brains to optimize the beam diameter of the excitation laser light and the value of the RI in the immersion liquid for the objective lens. Such optimizations allowed us to maximize the two-photon excitation efficiency by achieving tighter focusing in deeper regions of living mouse brains even with the conventional two-photon microscopy. These will also suggest insights regarding the functions of local neural networks that will be of further reference in the future.

## Materials and methods

### Ethics statement

All animal experiments were carried out following the recommendations in the Guidelines for the Care and Use of Laboratory Animals of the Animal Research Committee of Hokkaido University. All protocols were approved by the Committee on the Ethics of Animal Experiments at Hokkaido University (Permit Number: 17–0077). All surgery was performed under sodium pentobarbital anesthesia, and all efforts were made to minimize suffering.

### Animal specimens

Adult Thy1-eYFP (H-line) mice [24] of either sex (8–12 weeks of age) were used for all of the experiments, except for $Ca^{2+}$ imaging, where GLT1-G-CaMP7 (G7NG817-line) mice [25] of

either sex (8–12 weeks of age) were tested. All mice were housed with food and water *ad libitum* on a 12: 12 light-dark cycle (lights on from 8:00 to 20:00), with controlled temperature and humidity.

## Adjustments of RI value

The values of the RI in the agarose gels and the immersion liquids were adjusted to four decimal places, which is the precision limit of the refractometer (PAL-RI, ATAGO). Glycerol (Wako) was diluted with deionized water first, and then, the RI value was finely adjusted.

## Optical setup

All images were obtained by the two-photon laser scanning microscope system (FV1000 and BX-61WI, Olympus) with a 25× water immersion objective lens (NA 1.05, Olympus), which was equipped with a mode-locked Ti: Sapphire laser (Tsunami, Spectra-Physics). The excitation wavelength was 910 nm. To modify the beam diameters of the excitation laser light, a beam expander, a pair of convex lenses, was installed in the optical path. By changing the distance between lenses, the excitation light beam diameter at the position of the entrance pupil of the objective lens was modified from 10.8 (66.7% of the pupil diameter; underfill) to 17.4 mm (115.1% of the pupil diameter; overfill) (S1 Fig). The maximum value of laser power after the objective lens varied from 70 (overfill) to 165 mW (underfill). All fluorescence signals with the wavelength under 690 nm were detected via the multi-alkaline PMT (R7862, Hamamatsu). In the case of multicolor imaging of eYFP and red fluorescent beads, the fluorescence signal was divided at a wavelength of 570 nm by a dichroic mirror in front of the PMT.

## Specimen preparations for *in vitro* and *in vivo* evaluations of focal volumes

For *in vitro* measurements of focal volumes, 1% w/v agarose gel (agarose L, Nippon Gene), of which values of RI were modified by adding glycerol to the solvent was mixed with fluorescent nanobeads (200 nm yellow-green [505/515], Invitrogen) in the sub-diffraction-limited size to give enough fluorescent signals. The fluorescent nanobeads were observed under all combinations of the RI values in the immersion liquids (1.33, 1.34, 1.35, 1.36, and 1.37), depths (100, 400, and 800 μm), and excitation light beam diameters (overfill and underfill). To measure the focal volumes, we evaluated the full width at half maximums (FWHMs) of the fluorescence intensity profiles of the beads by fitting to the Gaussian distribution along the axial (z-axis) and lateral (x-axis) directions.

As in the *in vitro* case, we used a similar method for the *in vivo* evaluations of focal volumes. Fluorescent beads (200 nm crimson [625/645], Molecular Probes, or 1.0 μm red [580/605], Life Technologies) were dispersed in phosphate-buffered saline. A small hole was drilled in the mouse skull, and the beads were injected into the living mouse brain through a glass capillary by oil pressure. To reduce damages on the pathway of the excitation laser light, the glass capillary was inserted at a shallow angle (20˚–30˚) to the brain stereotaxic apparatus. No bleeding mice were used for the following image acquisitions. For the measurement of FWHMs, we presumed that the fluorescence intensity profiles follow a Gaussian distribution in both axial and lateral directions and the center pixels locate the peak of the Gaussian distribution. Such presumption was necessary to obtain both the axial and lateral intensity profiles given the blurry 3D images of single nanobeads observed from the effects of heartbeats, breathing and so on of the living mouse during cross-sectional image acquisitions.

### Cranial window implantations for *in vivo* two-photon imaging

To achieve high-resolution two-photon imaging of the living mouse brains, the overlying opaque skull bone must be partially removed to make a cranial window, the "open-skull" glass window [26]. In the open-skull preparation, a piece of the cranial bone (about 2.7 mm in diameter for the evaluations of the focal volumes in living mouse brains or 4.2 mm in diameter for the other experiments) was removed, but their dura was kept intact. The exposed brain was covered with a thin glass coverslip (No. S1, Matsunami) (for detailed methods, see published protocols [26]).

### Animal setup on the microscope stage

After the open-skull surgery, fluorescent beads (1.0 μm orange [540/560], Life Technologies) were affixed to the upper surface of the glass coverslip. Subsequently, the mice were placed on the microscope stage with an adapter plate under inhalation anesthesia. We ensured the horizontality of the coverslip by optimizing the angle of the adapter stage that has three adjuster bolts to reduce the effect of the optical aberrations arisen from a tilt. The angle of the adapter stage was manually controlled using the bolts and by observing the orange fluorescent beads as an indicator of the coverslip angle as we previously reported [18].

### Image acquisition and analysis

The square regions of the cortex (approximately 50–500 μm × 50–500 μm) and the regions of agarose gels (25 × 25 μm) were imaged at 512 × 512 pixels or 1024 × 1024 pixels at approximately 1–6 s per frame. In the case of fluorescent beads, it took 3 or 1 s per frame for 200-nm or 1.0-μm beads, respectively. Cross-sectional image stacks with 0.25 or 5.00 μm z-steps were acquired in the observations of fluorescent beads or living mouse brains, respectively. The background intensities were calculated from the mean values of each value of 2100 pixels in total, consisting of every 100 pixels in the order from the lowest intensities of 21 *xy*-planes. The signal intensities were calculated by subtracting the background intensities from the raw fluorescence intensities of the whole 21 *xy*-planes. The degree of contrasts of captured images was evaluated using the Brenner gradient [27]. For the extraction of the area where spontaneous calcium activities in astrocytes were observed, first, the background intensities obtained from regions of blood vessels were subtracted. Next, the lowest basal fluorescence intensities calculated from the minimum intensity projection image were subtracted at each pixel. Subsequently, the remaining intensities were binarized using Otsu's method. Finally, the foreground areas larger than 20 pixels were regarded as the region of interest.

### Focal laser ablation

The excitation laser light beam was repeatedly scanned (10 to $10^4$ times) along a 0.22-μm length line (corresponding to 2 pixels) with the maximum laser power. After the repetitive laser light irradiation, the laser power was lowered to the original (a few tens of mW) at the position of the specimens, and a voltage was applied to the PMT to restart image acquisitions.

### Software and statistics

All fluorescent images were acquired under control with FV10-ASW (Olympus). Evaluations of the focal volumes, imaging contrasts, fluorescence intensities and spontaneous $Ca^{2+}$ activities in astrocytes were conducted on Fiji (NIH). All z-stack images were reconstructed using a NIS element ver. 4.00 (Nikon). Data in the text and figures were shown as the mean ± standard error of the mean (s.e.m.). For the statistical analysis of the FWHMs and

made graphs, we used a software environment R. The comparisons of the FWHMs were conducted using Welch's t-test. For multiple comparisons, the Friedman test with Dwass–Steel–Critchlow–Fligner (DSCF) all-pairs comparison test was used.

## Results

The focal volumes are often visualized by single fluorescent nanobeads embedded in specimens. We visualized the focal volumes in living mouse brains by introducing sub-diffraction-limited fluorescent nanobeads (200 nm) into H-line mouse brains by stereotaxic microinjection (Fig 1A). We decided on the angle for the insertion by taking into account the diameter of the cranial window, maximum incident angle of the excitation light beam, and depths for the observation (Fig 1B). Because we inserted a glass pipette with a shallow angle of 20˚ to 30˚ with respect to the brain surface, the trace of the inserted glass pipette did not impair the excitation laser light beam from focusing ideally. We employed mice without any bleeding on the brain surface of the injection site as such bleeding would have impaired deep imaging of living mouse brains (Fig 1C). Two days after the injection, we carefully constructed the cranial window (φ2.7 mm) above the primary visual cortex and performed live imaging using the two-photon microscopy system. As a result, we succeeded in observing neurons and the injected red fluorescent beads simultaneously within living mouse brains (Fig 1D). Notably, the injection did not impair the maximum depth of two-photon observations nor induce dysmorphic

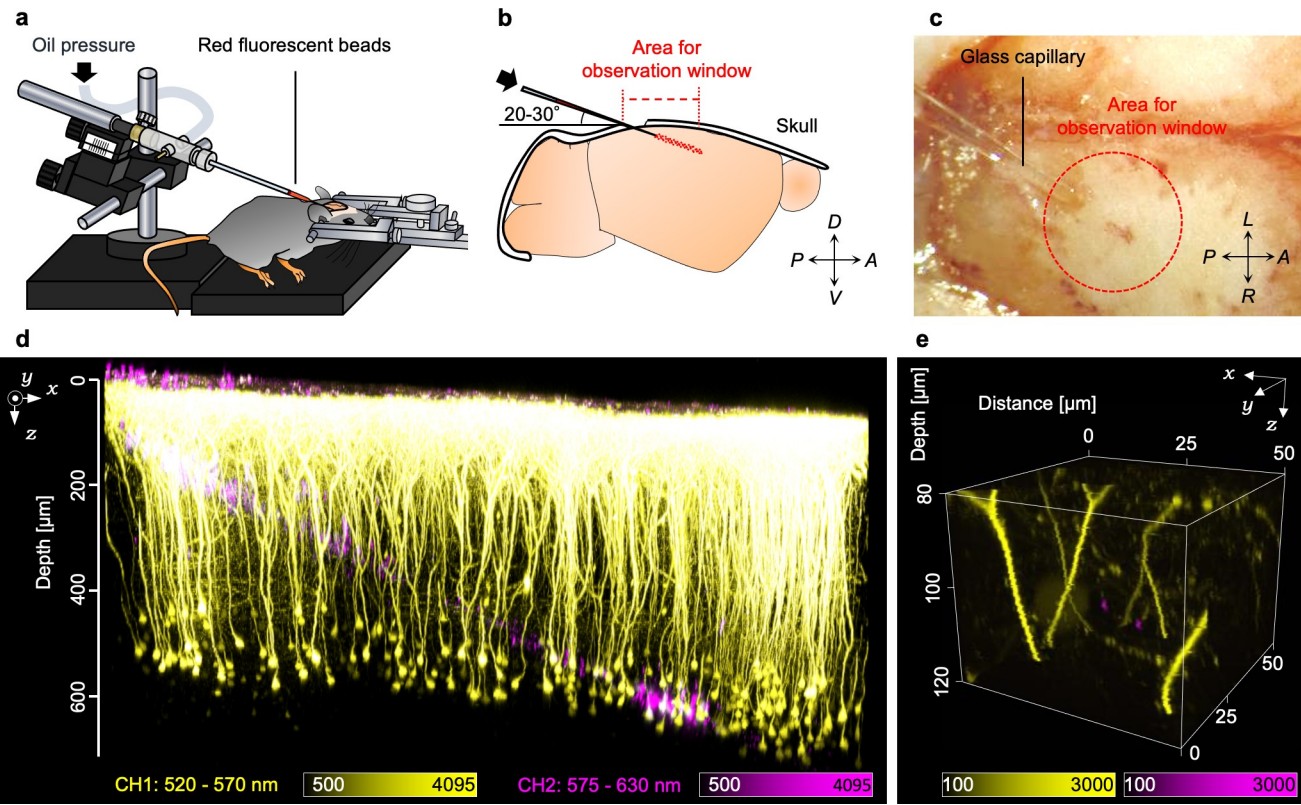

**Fig 1. Injection of fluorescent beads into living mouse brains.** (a) Schematic illustration of the injection method. The fluorescent red beads were injected into the living mouse brain by oil pressure through the glass capillary. (b) Schematic illustration of the injection strategy. The glass capillary was inserted into the brain at shallow angles of 20˚–30˚ to the brain stereotaxic apparatus. (c) A bright-field image of the skull during the injection. (d) A 3D reconstructed image of the neurons (yellow) and the injected red fluorescent beads (magenta). (e) An example of the 3D reconstructed image of the area containing two single fluorescent beads within the living mouse brain. A: anterior, P: posterior, D: dorsal, V: ventral, R: right, and L: left.

neural processes and somata, except for in the trace of the injection. In the trace and the surrounding area, many isolated single fluorescent nanobeads were observed from the brain surface to over 300-μm depth (Fig 1E). However, we could not observe single nanobeads to over 400-μm depth with our experimental conditions because of the low fluorescence intensity.

For the evaluations of the focal volumes in living mouse brains, we observed single fluorescent nanobeads in a 3D manner at the position of several depths (100, 200 or 300 μm from the brain surface). First, we captured several series of 3D fluorescent nanobeads with the excitation light beam diameter under the overfill condition using water as the immersion liquid (standard condition, condition S). The single nanobeads appeared to have expanded in volume on a depth-dependent manner, especially along the axial direction (*z*-axis) (Fig 2A). Next, we evaluated the depth dependency of FWHMs from the series of captured 3D images (*see Materials and methods*, Fig 2B). The shape of the averaged focal volume at the 300-μm depth ($FWHM_{xy}$ = 0.53 ± 0.03 μm, $FWHM_z$ = 4.14 ± 0.19 μm, n = 11 beads) was expanded to approximately 1.3 or 1.9 times along the lateral or axial directions, respectively, compared to the 100-μm depth ($FWHM_{xy}$ = 0.41 ± 0.01 μm, $FWHM_z$ = 2.16 ± 0.05 μm, n = 15 beads) and to the 200-μm depth ($FWHM_{xy}$ = 0.40 ± 0.01 μm, $FWHM_z$ = 2.33 ± 0.07 μm, n = 11 beads) (Fig 2C and 2D). Such depth-dependency of focal volume expansion, especially along the axial direction, indicated that spherical aberrations occurred predominantly, which was consistent with previous reports [28, 29].

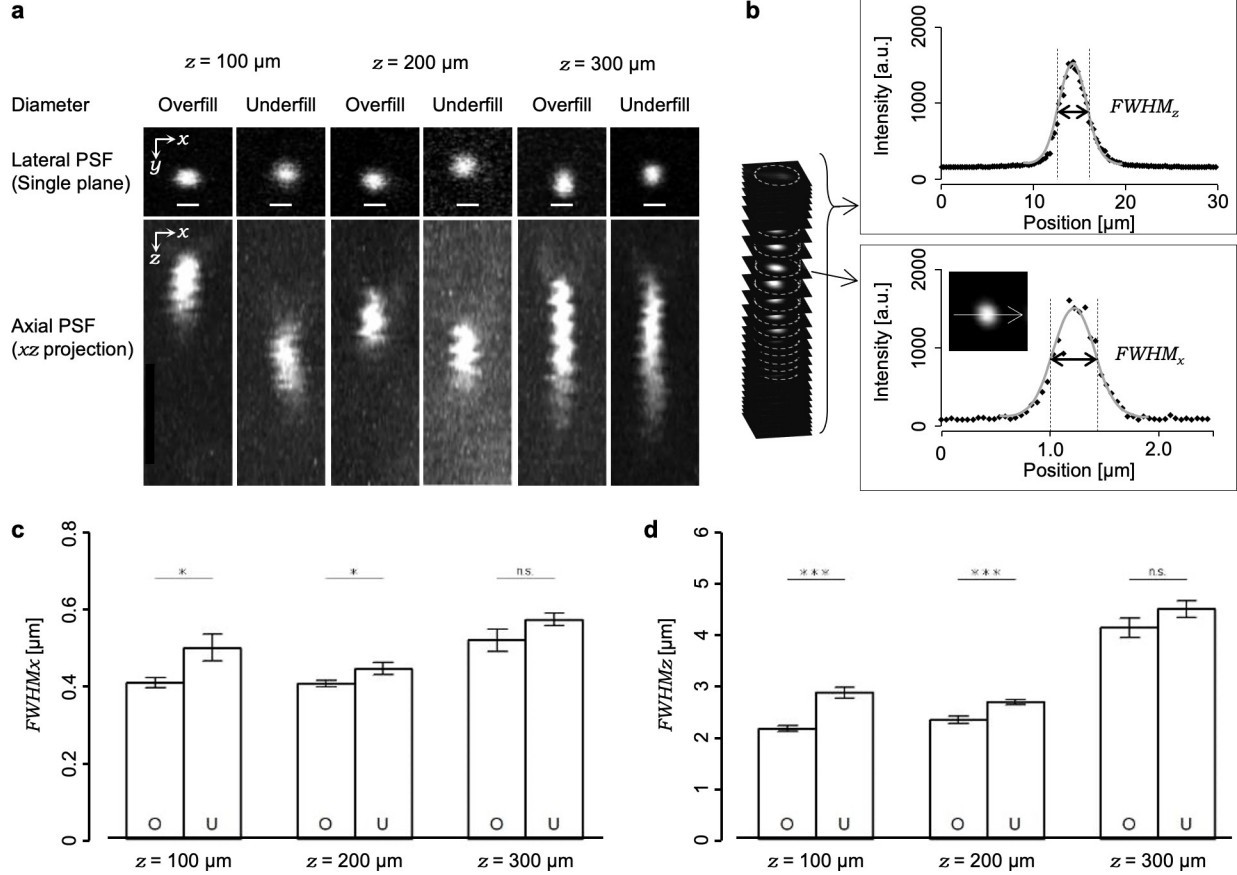

**Fig 2. Evaluations of focal volumes in living mouse brains under the overfill and underfill conditions.** (a) Examples of fluorescent nanobeads obtained under various conditions. The fluorescence intensities were normalized. All scale bars represent 1 μm. (b) Schematic illustration of the strategy for the measurements of the FWHMs along the axial and lateral directions. (c, d) Averaged lateral and axial FWHMs. O: overfill; U: underfill; *: p<0.05; ***: p<0.005 (Welch's t-test). Error bars represent s.e.m.

Although the optical theory tells us that the beam diameter of the excitation laser light should overfill the entrance pupil of the objective lens in order to utilize its full performance, it has been predicted that an appropriate reduction rate in the excitation light beam diameter could achieve higher excitation laser power without significant expansions of focal volumes in specimens because higher portions of the excitation light beam can pass through the entrance pupil [10, 30]. Thus, we examined and compared the effects of the reduction in the beam diameter of the excitation laser light with the overfill condition. As compared to the overfill condition, when the beam diameter of the excitation laser light was reduced to 67% of the pupil diameter of the objective lens (the underfill condition), the excitation laser power was doubled after the objective lens (S1B–S1D Fig). The focal volume reconstructed from 3D images of single fluorescent nanobeads showed depth-dependent expansion (100 μm, $FWHM_{xy} = 0.44 \pm 0.02$ μm, $FWHM_z = 2.68 \pm 0.05$ μm, n = 10 beads; 200 μm, $FWHM_{xy} = 0.50 \pm 0.03$ μm, $FWHM_z = 2.87 \pm 0.11$ μm, n = 10 beads; 300 μm, $FWHM_{xy} = 0.58 \pm 0.02$ μm, $FWHM_z = 4.51 \pm 0.19$ μm, n = 12 beads). Moreover, as compared to the overfill condition, the averaged focal volumes were significantly expanded by the reduction in the excitation light beam diameter in both depths of 100 and 200 μm. By contrast, we found that the averaged focal volumes in the 300-μm depth in the overfill and underfill conditions (Fig 2C and 2D) were not significantly different. This result is in accordance with the fact that the higher NA components of the excitation light beam hardly reached a deeper position of the specimens. Accordingly, the underfill beam diameter seemed suitable for deep imaging of living mouse brains while keeping focal volumes.One of the practical strategies proposed to contract the focal volume is reducing the RI mismatch between the immersion liquid and the biological specimens [31, 32]. Here, we first confirmed that the ideal spatial resolution under the cover-slip is kept in the 1.33–1.37 RI range in immersion liquids just by rotating the correction collar. Next, we employed two kinds of agarose-gel specimens with 200-nm, yellow–green fluorescent nanobeads embedded. In each specimen, the value of the RI was adjusted to 1.33 or 1.35. The focal volumes in the specimens were evaluated in three depth positions (100, 400, or 800 μm) having several RI values in the immersion liquid (1.33, 1.34, 1.35, 1.36, or 1.37) under two exci-tation light beam diameter conditions (overfill or underfill). The averaged focal volumes showed RI mismatch-dependent expansions especially in deep regions (S2 and S3 Figs). Nota-bly, using agarose-gel specimens with an RI value of 1.35, we confirmed that adjusting the RI of the immersion liquid contracted averaged focal volumes that comparing with the case of water used as the immersion liquid (S3 Fig).

Subsequently, using the method for evaluating the focal volume, we explored the optimal value of the RI in the immersion liquid to achieve a minimal focal volume in the living mouse cortex. In our experimental setup, the low fluorescence intensity of the single nanobeads pre-vented observations in positions deeper than 300 μm and evaluations of the focal volumes. To evaluate the focal volume in deeper regions under several conditions, we used the larger 1.0-μm diameter fluorescent microbeads, which showed a stable higher fluorescence intensity. Although the larger size was not suitable for evaluating focal volume accurately, it enabled us to evaluate the relative effects of RI modifications by measuring $FWHM_z$ because the 1.0-μm size was smaller than the measured $FWHM_z$ with nanobeads (Fig 2C). The larger-sized single fluorescent microbeads were successfully visualized at 400-μm depth, but it was difficult to observe them over 500-μm depth. The cross-sectional single fluorescent microbead images were acquired by all combinations of the RI values in the immersion liquid (1.33, 1.34, 1.35, 1.36, or 1.37), depths (100, 200, 300, or 400 μm), and the excitation light beam diameters (over-fill or underfill). Fig 3A shows examples of typical 3D reconstructed images of fluorescent microbeads obtained under the normal overfill condition. Note that the axial length of the microbead images seemed to vary with both values of RI and depths. To evaluate the effects of

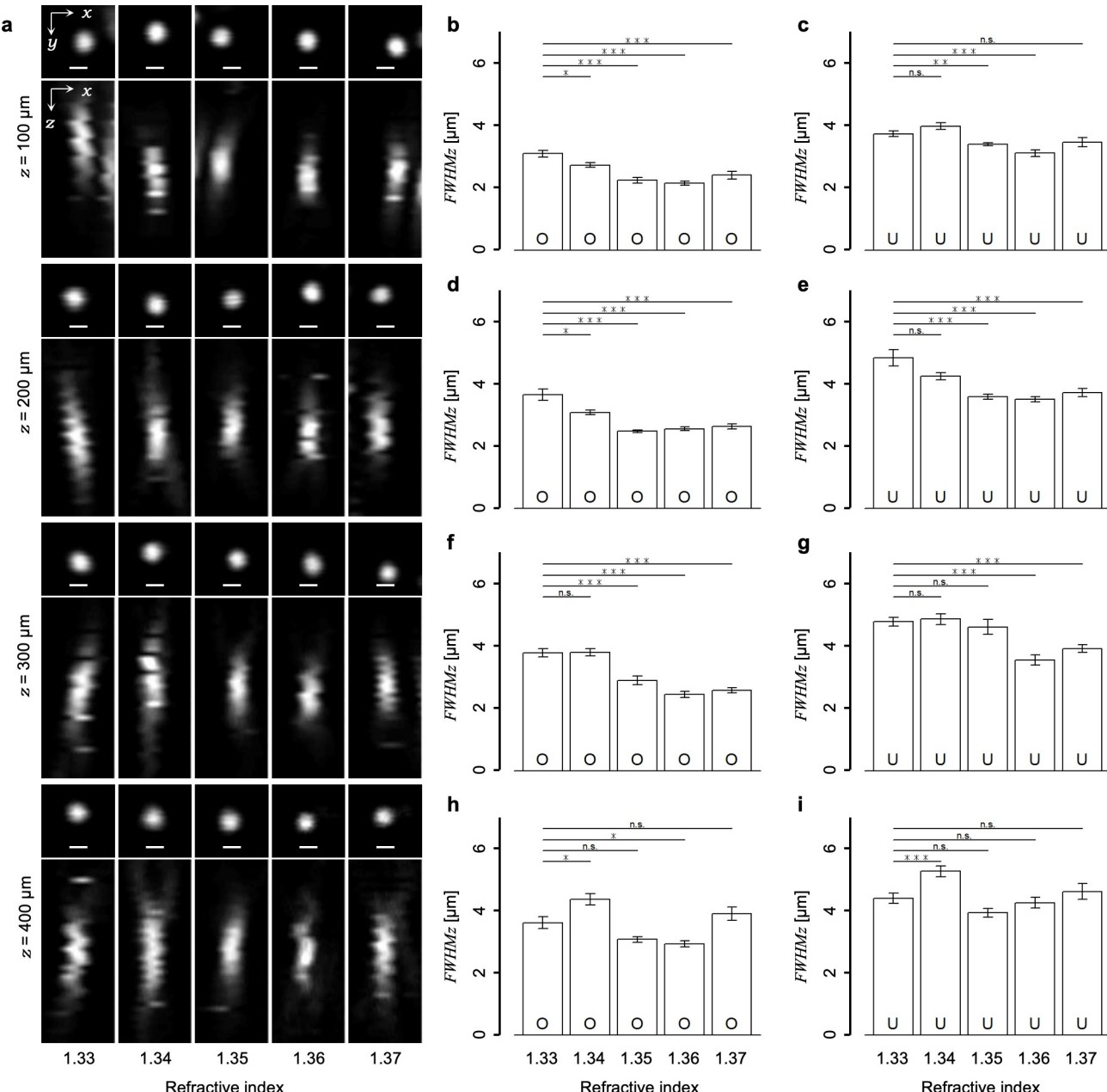

**Fig 3. Effects of the RI modulations on the focal volume in living mouse brains.** (a) Typical fluorescent microbead images obtained at each value of the RI in the immersion liquid (1.33, 1.34, 1.35, 1.36, and 1.37) and each depth (100, 200, 300, and 400 μm) under the overfill condition. The *x–y* images show single sections and the *x–z* images show max intensity projections. The fluorescence intensities were normalized. All scale bars represent 1 μm. (b–i) The averaged FWHMs measured from the intensity profiles of the bead images obtained under (b, d, f, h) the overfill and (c, e, g, i) underfill conditions. All statistical tests were carried out at the 1.33 condition with the other RI conditions. O: overfill; U: underfill; *: $p < 0.05$; **: $p < 0.01$; ***: $p < 0.005$ (Welch's t-test with Bonferroni correction). Error bars represent s.e.m.

RI modifications on the focal volumes, we measured the $FWHM_z$ from the axial intensity profiles of single fluorescent microbeads (*see Materials and methods*, Fig 2B). Consequently, the $FWHM_z$ varied significantly with the RI values as compared to that for the water immersion (n = 1.33) condition (Fig 3B–3I), but similarly with the results of the agarose gel experiments (S3 Fig). Apparently, the $FWHM_z$ was minimized with the adjustment of the RI in the

immersion liquid to 1.35 or 1.36 in most cases in the living mouse cortex (S1 Table). Therefore, we employed in the following experiments the immersion liquid with an RI of 1.36 to contract the focal volume in deep regions of living mouse brains.

Next, we examined the advantages of the modifications in the excitation light beam diameter and in the immersion liquid on the two-photon imaging of cortical layer V where single fluorescent microbeads could not be observed. The identical 3D region consisting of 21 cross-sectional *xy*-planes in the H-line mice was observed with the same average excitation laser power (70 mW, the maximum laser power under the overfill condition) under all combinations of the RI values (n = 1.33 or 1.36) and the excitation light beam diameters (overfill or underfill). The fluorescence intensities of the obtained images under n = 1.36 condition increased significantly as compared to the 1.33 condition (Fig 4A–4D). The enlarged views clearly showed the existence of a lot of finer structures like identical single dendritic spines in the same field of view under the 1.36 condition, as compared to that in the 1.33 condition (Fig 4E–4L). Moreover, we quantitatively evaluated the fluorescence intensities and contrasts in the images in the same 3D region using the Brenner gradient [27]. In Fig 4M and 4N, the values of the averaged signal intensities and Brenner gradients showed significant increases in the fluorescence intensities of neurons and in the contrasts under the 1.36 condition, as compared to under the 1.33 condition, demonstrating that the 1.36 condition was more suitable for the two-photon imaging of the deeper regions. This result likewise suggested that the contraction of the focal volume even in cortical layer V over 500-μm depth with both excitation light beam diameters. Furthermore, we examined the effects of the beam diameter in detail. Apparently, the fluorescence intensities under the underfill condition increased significantly as compared to that under the overfill condition (Fig 4M and 4N). In this setup, because we could increase the excitation laser power to 165 mW, the underfill condition was expected to enhance the fluorescence intensity. Therefore, we decided that the combination of the underfill condition and the 1.36 condition (condition E) was the most suitable among all combinations of beam diameters and RIs for the observation of cortical layer V and deeper regions in living mouse brains.

In order to demonstrate such an advantage of condition E for *in vivo* deep imaging, we observed adult H-line mice through the cranial window (φ4.2 mm) implanted above the primary visual cortex. Under condition S, all cortical layers were visualized, except for hippocampal neurons, affirming consistency with previous reports (Fig 5A) [11, 12]. By contrast, under condition E, not only hippocampi alveus but also hippocampal CA1 neurons were visualized with the normal multi-alkaline PMT detector (Fig 5B–5E). Notably, CA1 hippocampal neurons could not be observed without modifications of both the excitation light beam diameter and the value of the RI in the immersion liquid on the same mice (data not shown).

Additionally, we tested the advantages of condition E in the $Ca^{2+}$ imaging of living brains. We used G7NG817 mice [25] showing high expressions of a bright $Ca^{2+}$ indicator G-CaMP7 mainly in astrocytes and a small number of neurons. Spontaneous $Ca^{2+}$ activities of cortical layer V astrocytes in the primary visual cortex were observed under the anesthetized condition. The strong fluorescence signal of G-CaMP7 in G7NG817 mice allowed the observation of spontaneous $Ca^{2+}$ activities of cortical layer V astrocytes even under condition S. However, condition S showed very low contrasts and activity-dependent signal changes ($\Delta F$) (Fig 6A–6C), whereas condition E, having achieved not only smaller focal volumes but also higher excitation laser power after the objective lens, allowed the attainment of a high ratio of background-to-basal fluorescence intensity $F$ as well as obvious $\Delta F$ values (Fig 6D–6F). Nevertheless, distinguishing somata from the thin processes proved difficult with the dense labeling of G7NG817 mice. On the basis of the obtained dataset, we computationally extracted the area where a spontaneous $Ca^{2+}$ activity was observed as the time course of $\Delta F$ (*see*

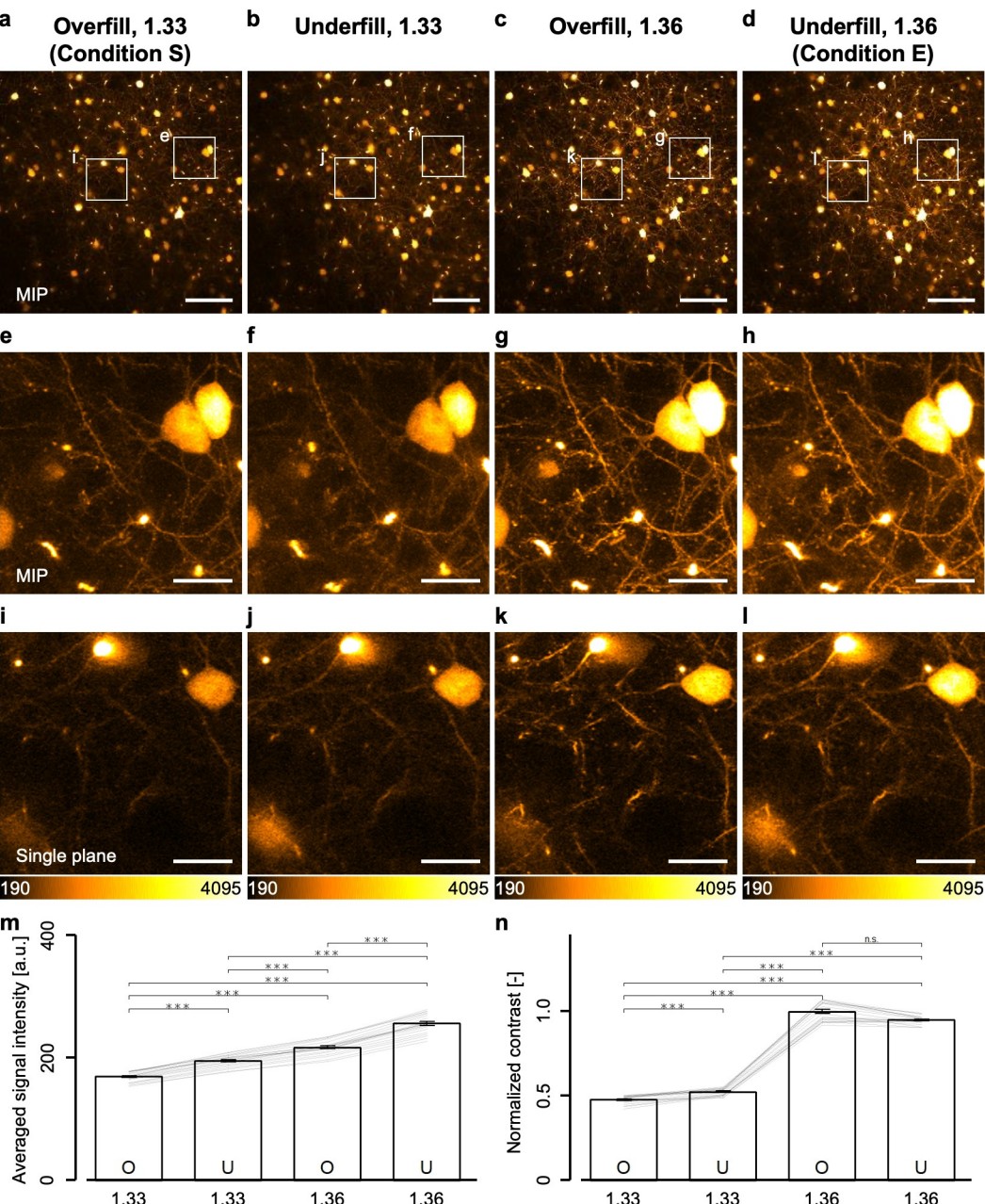

**Fig 4. Two-photon imaging of cortical layer V neurons in living mouse brains under all conditions.** (a-d) Max intensity projection images with 10 μm thickness obtained from cortical layer V under each condition. (e-h) Enlarged views in the white rectangle regions in (a–d). (i–l) An enlarged view in the white rectangle regions in (a–d) but not max intensity projections. (m) Averaged fluorescence signal intensities of 21 cross-sectional images under each condition. The lines in the graph show the calculated value of the signal intensities excluding the background intensities (*see Materials and methods*) at each *xy*-plane (21 cross-sectional images). (n) Contrast values normalized with the value of the combination of the overfill and 1.36 condition. The lines in the graph show the calculated value at each *xy*-plane (21 cross-sectional images). The scale bars shown in (a–d) and (e–l) represent 100 and 20 μm, respectively. Error bars represent s.e.m. ***p < 0.001 (Friedman test with post-hoc DSCF all-pairs comparison test).

*Materials and methods*). Various area sizes and maximum values of *ΔF* were detected with the unbiased computational extraction, which is consistent with previous reports [33, 34]. Moreover, the number of unbiased detection areas in the same field of view was not different in

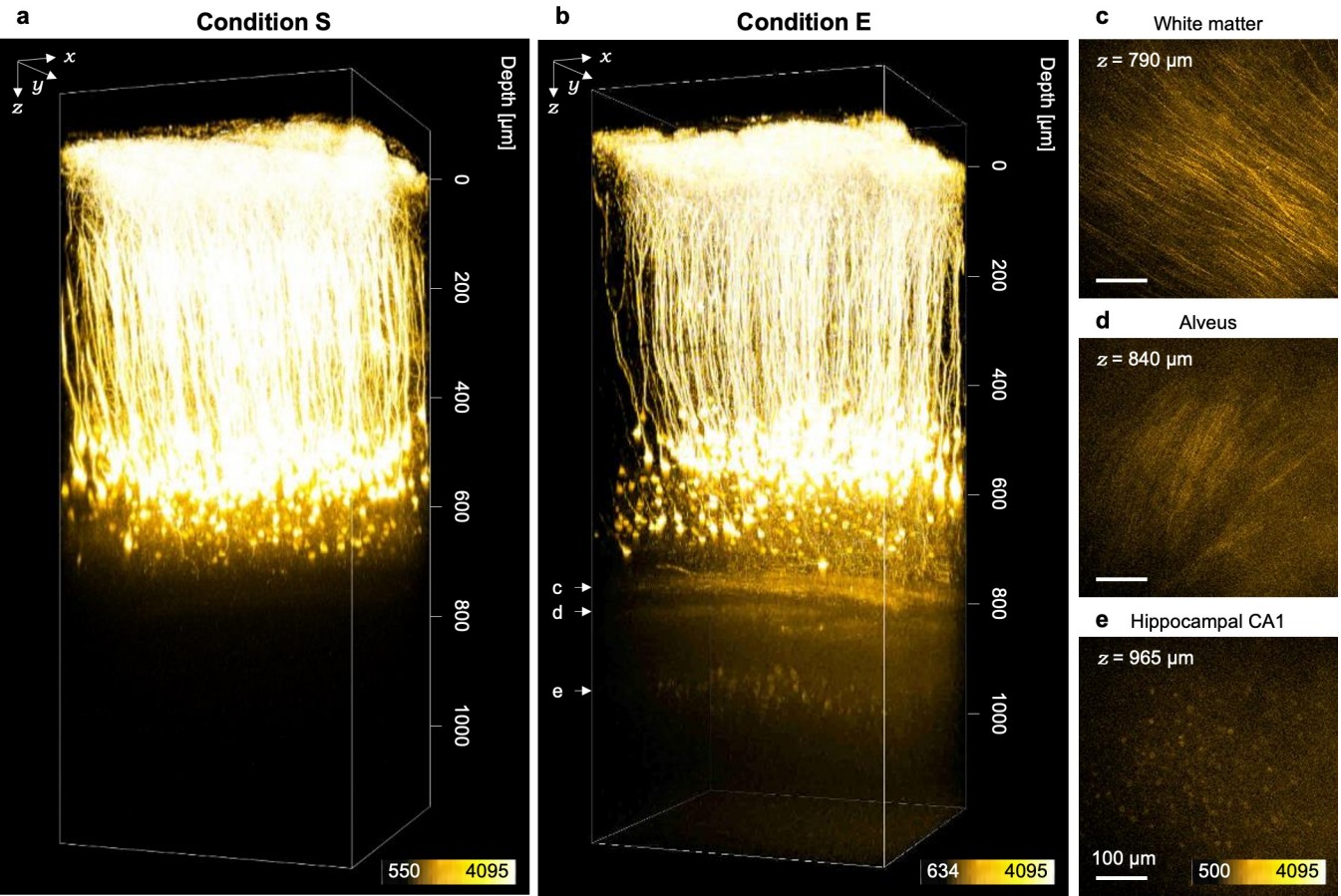

**Fig 5. Two-photon imaging of cortical and hippocampal neurons under condition E.** (a, b) 3D reconstructed images of a living mouse brain obtained under (a) condition S and (b) condition E. Hippocampal CA1 neurons were clearly visualized in (b). (c–e) Fluorescent images of the white matter, the alveus of hippocampi and hippocampal CA1 neurons, respectively.

both observation conditions, and no long-lasting abnormal activity in astrocyte populations was observed. These results indicated that the high excitation photon density achieved by condition E did not cause phototoxicity in the brain parenchyma [35]. Condition E improved *ΔF* to approximately four-fold higher than did condition S under the same specimen conditions (Fig 6B, 6C, 6E and 6F).

Increments of excitation photon density were achieved in deeper regions using the *in vivo* optical manipulation techniques. Finally, we demonstrated the focal laser ablation in living mouse brains with condition E by constantly irradiating the excitation laser light on selected points of neural processes at the maximum excitation laser power (70 mW for condition S and 165 mW for condition E) for 0.04–4 s. Such spatially restricted long-term irradiation precisely dissected the targeted neural processes in the living mouse brains (Fig 7A and 7B). Dissections were evaluated by long-lasting disappearances of the fluorescence signal in the irradiated area. Such disappearances can be distinguished from the photobleaching case because the fluorescence signals of the dissected dendrites did not recover within several minutes, whereas the signals would recover in the latter. Under condition S, which was similar to previously reported irradiation conditions [13, 14], dendrites were not dissected in depths of more than 300 μm corresponding to cortical layer II/III (Fig 7A and 7C). By contrast, condition E enabled the dissection of dendrites not only in cortical layer IV but also in layer V over 500-μm depth

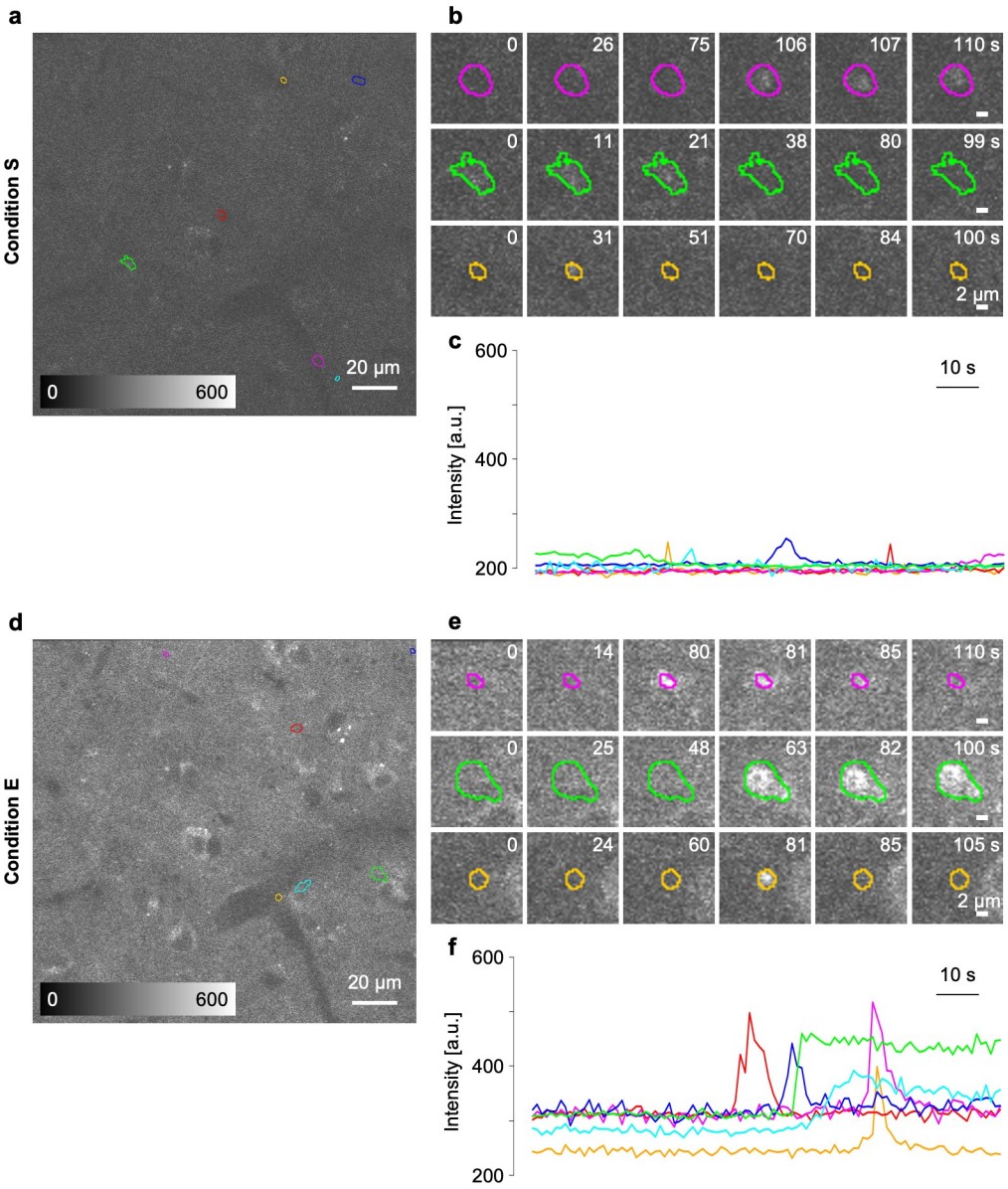

**Fig 6. Two-photon imaging of spontaneous Ca²⁺ activities of cortical layer V astrocytes under condition E.** (a, d) Fluorescence images of G7NG817-line mouse cortical layer V obtained in the same field of view under (a) condition S and (d) condition E. The colored region of interests (ROIs) show the activated area detected computationally. (b, e) Examples of dynamics of fluorescence intensity derived from the spontaneous activity in the automatically set ROIs. (c, f) The time course of fluorescence intensity in the ROIs. The scale bars in (a, c) and (b, d) represent 20 and 2 μm, respectively.

(Fig 7B and 7D). Surprisingly, we achieved the laser ablation of the single basal dendrites of layer V pyramidal neurons (Fig 7B). To investigate the effect of the focal laser ablation on both the targeted and non-targeted neurons, the same 3D area was observed the following day. The dissected apical dendrites showed degeneration over a length of 100 μm only along the distal direction. Nevertheless, the upper apical tufts maintained their original positions (n = 4 from two mice). In the closely located visible non-targeted neurons, their processes did not show any morphological distortions after the focal laser ablation (Fig 7D).

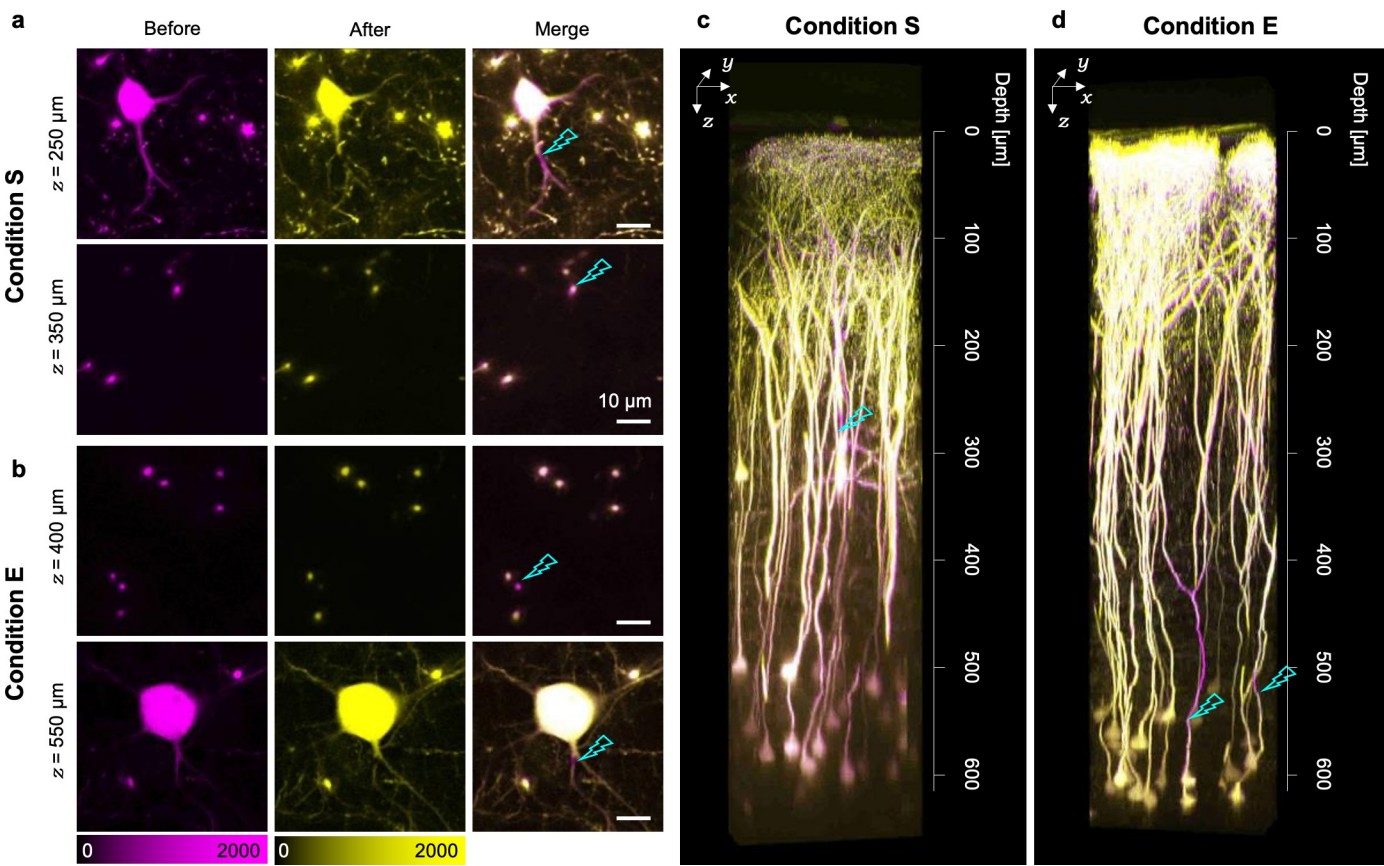

**Fig 7. Two-photon focal laser ablation in cortical layer V under condition E.** (a, b) Fluorescence images obtained before and after the laser ablation under (a) condition S and (b) condition E. The right panels show the merged images. All scale bars represent 10 μm. (c, d) 3D reconstructed images merged before focal laser irradiation and after 1 day under (c) condition S and (d) condition E. The magenta regions represent the sites disrupted by the focal laser ablation, and the cyan thunders represent the sites irradiated with laser light.

## Discussion

In this study, we developed a method for evaluating the focal volume in living mouse brains and successfully revealed a condition that maximizes the two-photon excitation efficiency by achieving tighter focusing especially in deeper regions (Figs 1–4; S2 Table; the underfill and 1.36 conditions). We also found that the combination of the overfill and 1.36 conditions was the best for *in vivo* imaging in shallow regions because it might achieve the minimum focal volume above cortical layer V (Fig 4; S2 Table). The maximized two-photon excitation efficiency obviously visualized hippocampal CA1 neurons (Fig 5) and Ca²⁺ activities in cortical layer V astrocytes (Fig 6). Noticeably, it enabled the focal laser ablation of both a single basal dendrite and a single apical one of cortical layer V pyramidal neurons (Fig 7).

Recently, focal laser ablations have been employed in investigating the contributions of single cells or single dendrites to sensory representation or orientation tuning, respectively [36, 37]. In these studies, focal laser ablations were not performed deeper than cortical layer II/III because of the limitation of the maximum depth [13, 14], which prevented detailed investigations of the functions of excitatory cortico–cortical connections across various cortical layers [38, 39]. The focal laser ablation in cortical layer V that we have achieved in this study (Fig 7) would allow us to evaluate the contributions of single cells or single dendrites to the functions of local cortical networks, which would follow the successful dissection of intercellular

connections in local cortical networks. Moreover, such tighter focusing would assist investigations of not only neuron–neuron interactions but also layer-specific neuron–astrocyte interactions [40] as it allows $Ca^{2+}$ imaging of astrocytes in deeper regions such as cortical layer V (Fig 6). In addition, the contraction of the focal volume (Fig 4) would improve the precision of other nonlinear optical manipulation techniques such as photo-uncaging [41–43] and optogenetics [44] via the localization of the photoactivatable area. As a future direction of this research, we could use such high–precision optical manipulation techniques to investigate in detail the contributions of single dendrites or single dendritic spines on information processing in living brains.

In Fig 3, the immersion liquid with an RI value of 1.35 or 1.36 minimized the focal volume in deep regions of the living mouse cortex mainly by compensation of the spherical aberration caused by the specimens (S2 Fig; S1 Table) [21, 23, 31]. The minimized focal volume indicated that the RI mismatch between the immersion liquid and the specimens might also be reduced. The result suggests that the average value of the RI in the living mouse cortex ranges from 1.35 to 1.36, which is consistent with a result from a recent report [23]. This consistency also suggests that the method developed here may evaluate the average value of the RI in various biological specimens.

It was suggested that the simple modification in the immersion liquid compensated the dominant part of spherical aberrations owing to the similar minimization of the focal volume in optical phantoms such as the agarose gel specimen under the ideal condition (S2 and S3 Figs). The residual optical aberrations that could not be compensated in living mouse brains (Fig 3) might be caused by their complicated structures composed of various elements with a specific value of the RI such as blood vessels, white matter, extracellular matrix, and intracellular organelles [45, 46]. In this sense, other sophisticated AO techniques would be required to completely compensate for optical aberrations in living mouse brains [19–22].

In this study, the tighter focusing that minimized the focal volume and maximized the two-photon excitation efficiency was achieved especially in deeper regions of living mouse brains by suppressing optical aberrations or scattering. Such tighter focusing successfully demonstrated $Ca^{2+}$ imaging and focal laser ablation in deeper regions with the conventional two-photon microscopy system. Of course, the procedures for the tighter focusing is compatible with other established technologies such as high-peak-power [18, 47], longer-wavelength laser light sources [47, 48], high-sensitivity detectors [18, 49], AO devices [19–23], and various fluorescent probes [50–52]. The novel methodology developed here could enable a more precise investigation of intercellular communications.

## Supporting information

**S1 Fig. Modulation in the diameter of the excitation laser light beam.** (a) Schematic illustration of the optical setup of the two-photon microscopy. (b) A graph showing the relationship between the position of the movable lens and the excitation laser power after the objective lens. (c, d) Measured intensity profiles of the excitation laser light beam before the pupil under (c) the overfilled condition and (d) the underfilled condition.
(TIF)

**S2 Fig. Evaluations of focal volumes in the n = 1.33 agarose gel specimen.** (a–h) Averaged FWHMs evaluated from the fluorescence intensity profiles of the beads embedded in the agarose-gels of which RIs were indicated as the white bar under each condition. All statistical tests were carried out at the RIs of the agarose gels with the other RI conditions. $^*$: $p < 0.05$; $^{**}$: $p < 0.01$; $^{***}$: $p < 0.005$ (Welch's t-test with Bonferroni correction). Error bars represent s.e.

m.
(TIF)

**S3 Fig. Evaluations of focal volumes in the n = 1.35 agarose gel specimen.** (a–h) Averaged FWHMs evaluated from the fluorescence intensity profiles of the beads embedded in the agarose-gels of which RIs were indicated as the white bar under each condition. All statistical tests were carried out at the RIs of agarose-gels with the other RI conditions. *: $p < 0.05$; **: $p < 0.01$; ***: $p < 0.005$ (Welch's t-test with Bonferroni correction). Error bars represent s.e. m.
(TIF)

**S1 Table. FWHMz evaluated under each condition.** FWHMz measured from intensity profiles of single microbeads obtained under each condition. All values represent mean ± s.e.m. Numbers in the parentheses represent the number of evaluated beads. Cells of gray background show the minimum FWHMs at each depth.
(TIF)

**S2 Table. Summary of Summary of the imaging condition investigated in this study.** Double circle, excellent; circle, good; triangle, average; cross, bad.
(TIF)

## Acknowledgments

We would like to thank Dr. Hajime Hirase of RIKEN CBS for providing us the G7NG817-line. We are grateful to Dr. H. Ujii of the Graduate school of information science and technology, Hokkaido University, for his valuable comments and Dr. R. Enoki, Dr. K. Iijima, Dr. H. Ishii, Dr. M. Tsutsumi of Research Institute for Electronic Science, Hokkaido University, Dr. L. Qiao of the department of Health Sciences, Hokkaido University, for their constructive suggestions. We are also grateful to Dr. K. Kobayashi, and Dr. Y. Matsuo of the Nikon Imaging Center at Hokkaido University for providing technical support. We would like to thank Enago (www.enago.jp) for the English language review.

## Author Contributions

**Conceptualization:** Ryoji Kitamura, Ryosuke Kawakami, Kohei Otomo, Tomomi Nemoto.

**Data curation:** Ryoji Kitamura, Ryosuke Kawakami, Kohei Otomo, Tomomi Nemoto.

**Formal analysis:** Kazushi Yamaguchi, Ryoji Kitamura.

**Funding acquisition:** Ryosuke Kawakami, Tomomi Nemoto.

**Investigation:** Kazushi Yamaguchi, Ryoji Kitamura.

**Methodology:** Kazushi Yamaguchi, Ryoji Kitamura, Ryosuke Kawakami, Tomomi Nemoto.

**Project administration:** Ryosuke Kawakami, Kohei Otomo, Tomomi Nemoto.

**Resources:** Kazushi Yamaguchi, Ryoji Kitamura, Ryosuke Kawakami, Kohei Otomo, Tomomi Nemoto.

**Software:** Kazushi Yamaguchi, Ryoji Kitamura.

**Supervision:** Tomomi Nemoto.

**Validation:** Kazushi Yamaguchi, Ryoji Kitamura.

**Visualization:** Kazushi Yamaguchi, Ryoji Kitamura, Ryosuke Kawakami, Kohei Otomo, Tomomi Nemoto.

**Writing – original draft:** Kazushi Yamaguchi, Ryosuke Kawakami, Kohei Otomo, Tomomi Nemoto.

**Writing – review & editing:** Kazushi Yamaguchi, Ryoji Kitamura, Ryosuke Kawakami, Kohei Otomo, Tomomi Nemoto.

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
