## [Decision Letter · Decision Letter 0]

23 Jul 2020

In vivo two-photon microscopic observation and ablation in deeper brain regions realized by modifications of excitation beam diameter and immersion liquid

PONE-D-20-15214

Dear Dr. Nemoto,

We’re pleased to inform you that your manuscript has been judged scientifically suitable for publication and will be formally accepted for publication once it meets all outstanding technical requirements.

Kind regards,

Lawrence B Cohen

Academic Editor

PLOS ONE

1. To comply with PLOS ONE submissions requirements, please provide methods of sacrifice in the Methods section of your manuscript.

Please amend your manuscript and send a revised copy by return email and we will upload this on your behalf

"No"

Please provide an amended Funding Statement that declares *all* the funding or sources of support received during this specific study (whether external or internal to your organization) as detailed online in our guide for authors at http://journals.plos.org/plosone/s/submit-now. 

Please state what role the funders took in the study.  If any authors received a salary from any of your funders, please state which authors and which funder. If the funders had no role, please state: "The funders had no role in study design, data collection and analysis, decision to publish, or preparation of the manuscript."

Please send your amended statements by return email; we will change the online submission form on your behalf.

Reviewers' comments:

Reviewer's Responses to Questions

**Comments to the Author**

1. Is the manuscript technically sound, and do the data support the conclusions?

Reviewer #1: Yes

Reviewer #2: Yes

2. Has the statistical analysis been performed appropriately and rigorously? 

Reviewer #1: Yes

Reviewer #2: Yes

3. Have the authors made all data underlying the findings in their manuscript fully available?

Reviewer #1: Yes

Reviewer #2: Yes

4. Is the manuscript presented in an intelligible fashion and written in standard English?

Reviewer #1: Yes

Reviewer #2: Yes

5. Review Comments to the Author

Reviewer #1: I recommend publication of the manuscript “In vivo two-photon microscopic observation and ablation in deeper brain regions realized by modifications of excitation beam diameter and immersion liquid.”

The experiments were carried out using sound methods and technology. The statistical analysis was performed properly.

The authors developed the methods that enable/improve visualization of deeper brain layer and the finer structure of hypocampal CA1 neurons. It is very interesting that those results were achieved by modifying experimental conditions using conventional two-photon microscopy. The data support the conclusion of the manuscript.

Reviewer #2: I was impressed by the manuscript “In vivo two-photon microscopic observation and ablation in deeper brain regions realized by modifications of excitation beam diameter and immersion liquid.” by Dr. Tomomi Nemoto and colleagues.

It seemed to me that the experiments were carefully and comprehensively carried out. The suggested experimental changes, under-filling the objective and increasing the refractive index had impressive and important improvements in spatial resolution, depth, and signal size. I recommend publication.

The English was fine but not always perfect. Perhaps an Editor can have a look.

6. PLOS authors have the option to publish the peer review history of their article (what does this mean?). If published, this will include your full peer review and any attached files.

Reviewer #1: **Yes: **Yunsook Choi

Reviewer #2: **Yes: **Lawrence B. Cohen

---

## [Editor Report · Acceptance letter]

29 Jul 2020

PONE-D-20-15214 

In vivo two-photon microscopic observation and ablation in deeper brain regions realized by modifications of excitation beam diameter and immersion liquid 

Dear Dr. Nemoto:

I'm pleased to inform you that your manuscript has been deemed suitable for publication in PLOS ONE. Congratulations! Your manuscript is now with our production department. 

Kind regards, 

on behalf of

Dr. Lawrence B Cohen 

Academic Editor

PLOS ONE